# Factors Affecting the Stability of Platinum(II) Complexes with 1,2,4-Triazolo[1,5-*a*]pyrimidine Derivatives and Tetrahydrothiophene-1-Oxide or Diphenyl Sulfoxide

**DOI:** 10.3390/ijms23073656

**Published:** 2022-03-26

**Authors:** Mateusz Jakubowski, Iwona Łakomska, Adriana Kaszuba, Andrzej Wojtczak, Jerzy Sitkowski, Andrzej A. Jarzęcki

**Affiliations:** 1Faculty of Chemistry, Nicolaus Copernicus University in Toruń, 7, 87-100 Toruń, Poland; mjakubowski@umk.pl (M.J.); akaszuba@doktorant.umk.pl (A.K.); awojt@chem.umk.pl (A.W.); 2National Institutes of Medicines, 30/34, 00-725 Warszawa, Poland; j.sitkowski@nil.gov.pl; 3Institutes of Organic Chemistry, Polish Academic of Science, 44/52, 01-224 Warszawa, Poland; 4Department of Chemistry, Brooklyn College of the City University of New York, New York, NY 11210, USA; jarzecki@brooklyn.cuny.edu; 5Ph.D. Program in Chemistry and Ph.D. Program in Biochemistry, The Graduate Center of the City University of New York, New York, NY 10016, USA

**Keywords:** platinum(II) complex, triazolopyrimidine, NMR, DFT, cis-/trans isomers

## Abstract

The platinum(II) complexes of general formula [PtCl_2_(dstp)(S-donor)] were dstp 5,7-dimethyl-1,2,4-triazolo[1,5-*a*]-pyrimidine (dmtp), 5,7-ditertbutyl-1,2,4-triazolo[1,5-*a*]pyrimidine (dbtp), 5-methyl-7-isobutyl-1,2,4-triazolo[1,5-*a*]pyrimidine (ibmtp) or 5,7-diphenyl-1,2,4-triazolo[1,5-*a*]pyrimidine (dptp), whereas S-tetrahydrothio-phene-1-oxide (TMSO) or diphenyl sulfoxide (DPSO) were synthesized in a one-pot reaction. Here, we present experimental data (^1^H, ^13^C, ^15^N, ^195^Pt NMR, IR, X-ray) combined with density functional theory (DFT) computations to support and characterize structure–spectra relationships and determine the geometry of dichloride platinum(II) complexes with selected triazolopyrimidines and sulfoxides. Based on the experimental and theoretical data, factors affecting the stability of platinum(II) complexes have been determined.

## 1. Introduction

Coordination compounds of transition metals having a d^8^ configuration such as Rh(I), Pd(II), Au(III), as well as Pt(II) exhibit square-planar geometry [1,2]. Most synthetic pathways that achieve this type of platinum(II) complexes involve associative ligand substitution [3]. These [PtX_2_N_2_] complexes, where X is a halide and N is an N-donor ligand, can exist as two *cis* or *trans* isomers, displaying different properties and antitumor activities [4]. Stereochemistry of the final product results from the *trans* effect of ligands within a complex [3,5].

Cisplatin (*cis-*[PtCl_2_(NH_3_)_2_]) is a worldwide platinum(II) anticancer drug currently used in treating various types of tumors. However, isomer *trans* (transplatin (*trans-*[PtCl_2_(NH_3_)_2_]) is inactive against tumor cell lines. Those simple examples confirm that the position of the ligand in the coordination sphere creates specific properties such as cytotoxicity [6]. Determining a specific isomer of platinum(II) compounds ([PtX_2_N_2_]) is significant for their structural and future application, especially antitumor studies. Usually, the isomers are distinguished by X-ray experiments for single crystals, IR spectra, or the Kurnakov test [7]. The Kurnakov reaction determines the *cis*/*trans* PtCl_2_N_2_ isomers upon treatment with an excess of thiourea (tu), giving different solubility products. *Trans* isomer converts into the white, insoluble powder of *trans*-[Pt(N)_2_(tu)_2_]Cl_2_, whereas cisplatin converts into yellow, soluble complex [Pt(tu)_4_]Cl_2_.

In general, the two isomers could be distinguished by NMR spectroscopy. However, the effectiveness and application of the method strongly depend on the ligands. For example, *cis*-[PtCl_2_(NH_3_)_2_] and *trans-*[PtCl_2_(NH_3_)_2_] structures are difficult to distinguish by ^195^Pt NMR since their chemical shifts are very alike (δ −2104 and −2101 ppm, respectively) [8]. However, the ranges of chemical shifts for dichloride platinum(II) complexes with pyridine derivatives differentiate enough to identify the two isomers. In these cases, the *cis* chemical shifts are observed between −1998 and −2021 ppm, while the *trans* shifts are between −1948 and −1973 ppm [9].

Our previous research on dichloride platinum(II) compounds with 1,2,4-triazolo[1,5-*a*]pyrimidine derivatives (dstp) led to obtain series of *cis*-[PtCl_2_(dstp)_2_] [10,11], *cis-*[PtCl_2_(NH_3_)(dstp)] [10,11] and *trans*-[PtCl_2_(dstp)(DMSO)] [11,12] complexes. Infrared studies confirmed the stereochemistry of those complexes. According to the group theory, IR spectra for *cis* complexes revealed two stretching vibrations for Pt-Cl (in the range 329–342 cm^−1^) and two absorption bands assigned to Pt-N stretching vibrations (in the range 280–287 cm^−1^) [10,13].

Here, as a continuation of a study in this field, we present experimental data (^1^H, ^13^C, ^15^N, ^195^Pt NMR, IR, X-ray) combined with density functional theory (DFT) computations to support and characterize structure–spectra relationships and determine the geometry of dichloride platinum(II) complexes with N-donor ligands. We selected series of novel dichloride platinum(II) complexes with disubstituted 1,2,4-triazolo[1,5-*a*]pyrimidine (dstp) derivatives and two different sulfoxides. The first series (**A**) is four platinum(II) compounds with tetrahydrothiophene-1-oxide (TMSO) with a general formula, *cis-*[PtCl_2_(dstp) (TMSO)], and the second (**B**) is another four complexes with diphenyl sulfoxide (DPSO) of a general formula, *trans-*[PtCl_2_(dstp) (DPSO)], where dstp ligands in the series are: 5,7-dimethyl-1,2,4-triazolo[1,5-*a*]pyrimidine (dmtp), 5,7-ditertbutyl-1,2,4-triazolo[1,5-*a*]-pyrimidine (dbtp), 5-methyl-7-isobutyl-1,2,4-triazolo[1,5-*a*]pyrimidine (ibmtp) or 5,7-diphenyl-1,2,4-triazolo[1,5-*a*]pyrimidine (dptp) (Figure 1). Structural studies incorporate X-ray diffraction experiments (solid-state) and multinuclear NMR spectroscopy (solutions). Additionally, to study structural differences between *cis/trans* isomers, we have compared experimental, and DFT calculated values of ^1^H, ^13^C, ^15^N, and ^195^Pt NMR chemical shifts, bond lengths, and angles for eight novel platinum(II) complexes with TMSO or DPSO. We complemented the study by computing the DFT energy scan of the relative rotation of the N-donor ligand and a sulfoxide and its potential impact on the relative stabilization of the *cis/trans* isomers for both series of (**A**) and (**B**).

Based on the good agreement of experimental results with DFT calculations, we can suggest spectroscopic techniques as a helpful method for predicting the *cis*/*trans* isomerism of platinum(II) complexes without the crystal structure data. This is a continuation of a long-standing investigation of our research group in order to understand the stability and biological function of platinum(II) complexes with triazolopyrimidines for medical and pharmacological applications.

## 2. Results and Discussion

### 2.1. Crystal Structures of cis-[PtCl_2_(dmtp)(TMSO)] (**A1**), cis-[PtCl_2_(dbtp)(TMSO)] (**A3**), and trans-[PtCl_2_(dbtp)(DPSO)] (**B3**)

The selected bond lengths and angles for (**A1**, **A3**, **B3**) are given in Table 1, whereas crystal data are presented in the experimental part.

In the reported complexes, the coordination sphere of Pt(II) has a square-planar geometry (Figure 2). Complexes (**A1**) and (**A3**) containing the disubstituted triazolopyrimidine ligand (dstp) and cyclic TMSO have been obtained as *cis-*dichloride isomers while (**B3**) is the *trans*-dichloride isomer. In (**A1**) and (**A3**), the valence geometry of the coordination sphere is almost identical (Table 1). In both complexes, the shortest bond is with the tp N3 atom, and both Pt-Cl bonds are slightly longer than the Pt-S bonds. Analysis of (**A1**) and (**A3**) reveals that the *trans* effect of the TMSO S-donor atom is stronger than that of N3 since Pt1-Cl2 in (**A1**) and Pt1-Cl1 in (**A3**) *trans* to Pt-S are longer than the other Pt-Cl bonds in both complexes. In the *trans*-dichloride isomer found for (**B3**), both Pt-Cl bonds are significantly shorter than those in complexes (**A1**) and (**A3** A significant difference in the Pt-Cl distances found in (**B3**) can be attributed to the Cl2 position, as it is more crowded with the phenyl ring of DPSO and the ring system of dbtp. In contrast, the Cl1 atom is away from the rings of DPSO. Moreover, in (**B3**), the Pt1-S1 distance of 2.2150(18) Å is significantly longer than that of 2.199 Å found in (**A1**, **A3**). That reflects the significant steric effect of the DPSO ligand in (**B3**). It is supported by comparing the valence angles of the coordination sphere with both N3-Pt1-Cl angles. These angles notably deviate from 90°. In (**B3**), the angles decrease by up to 2.4° while both angles of S1-Pt1-Cl increase up to 2.9°. However, in (**A1**) and (**A3**), the angle deviations are less than 1.6° (Table 1). 

The superposition of molecules (**A1**) and (**A3**) based on atoms of the coordination sphere PtSNCl_2_ (Figure 3) reveals the identical orientation of the TMSO ring but the opposite of the triazolopyrimidines moiety. Due to the heteroatom positions in 1,2,4-triazolopyrimidines, the partial charge at C2 and C3a might be positive. In (**A1**), the C3a of dmtp is 3.016 Å away from the TPSO O1, and a 3.37 Å distance was found between H2(C2) and Cl2. In the dbtp complex (**A3**), similar intramolecular contacts are also found (C3a…O1 and H2…Cl1, with 3.124 Å and 3.707 Å, respectively). Additionally, the t-Bu substituent at C5 is in the cavity of the coordination plane, the C21 methylene, and the S = O group. Therefore, the resulting tilt of the tp ring system relative to the coordination plane is different, with the dihedral angle between the best planes of 66.52(14)° in (**A1**) and 77.35(19)° in (**A3**). The comparison reveals a slight difference in the position of the TMSO S atom relative to the PtCl_2_N best plane. In (**A3**), the displacement of S from that plane is 0.043 Å, while in (**A1**), the distance is 0.190 Å.

In (**B3**), the DPSO ligand is in such an orientation that its O atom is close to the PtCl_2_NS coordination plane, with the Cl2-Pt1-S1-O1 torsional angle 24.33°. In such a position, the DPSO C31–C36 phenyl ring fits the cavity between C53 and C54 methyl groups of the t-Bu substituent. The intramolecular H…H interactions involving C53, C54 methyl groups of t-Bu and C35, C36 groups of DPSO are detected, with the distances 2.682 and 3.345 Å. The dihedral angle between the coordination plane and the triazolopyrimidine ring plane is 70.62°.

Comparison of the dbtp complexes (**A3**) and (**B3**) revealed that Pt1-N3-C2 angles are 129.7(5)° and 134.9(5)°, while Pt1-N3-C3A angles are 125.8(5)° and 123.2(4)° in (**A3**) and (**B3**), respectively. These data suggest the significant dispersion attractive interactions between t-Bu groups of dbtp and phenyl moiety of DPSO and to a certain extent with methylene groups of TMSO. On the other hand, in (**A1**), the less bulky methyl substituents in dmtp do not form any strong interactions with the methylene groups of TMSO, so they do not affect the dmtp position, and the resulting angles Pt1-N3-C2 of 127.3(3)° Pt1-N3-C3A 127.3(3)° are identical. The most substantial effect found for (**B3**) is confirmed by the S1-Pt1-N3 angle of 175.29(16)°, while in (**A1**) and (**A3**) the *trans* positioned Cl ligands to form N3-Pt1-Cl angles of 178.63(11)° and 179.17(19)°, respectively, much closer to the symmetric 180°.

Based on the architecture of (**A1**, **A3**, **B3**), the observed preferences for the complex isomers are supported by intramolecular interactions within the opposite isomers containing TMSO and DPSO. For hypothetic *trans*-[PtCl_2_(dstp)(TMSO)], the lack of significant interactions stabilizing the molecule might be expected between the TMSO methylenes and triazolopyrimidines. Therefore, that might be a discriminating factor responsible for the observed preference of *cis* isomers. For hypothetical *cis-*[PtCl_2_(dstp)(DPSO)], the steric hindrance generated by the ring systems of dstp and DPSO might be the factor explaining why isomers are unfavorable. That hypothesis is consistent with the potential energy surfaces calculated with DFT and reported further in the paper. It is worth pointing out that the global minima conformations computed by DFT correspond to those found in the crystal structures reported here.

Complete geometry optimization performed with DFT calculation for compounds (**A1**), (**A3**), and (**B3**) gave results in good agreement with those obtained by X-ray diffraction studies concerning the bond lengths and angles (Table 1). Additionally, a correlation of experimental data with theoretical calculation allowed us to compute structures and DFT energies for complexes we do not have isolated signal crystals for (Table 2). Based on DFT calculations, we can conclude that all *cis* isomers with TMSO are energetically favored, whereas all structures with DPSO tend towards *trans* configuration.

### 2.2. Multinuclear ^1^H, ^13^C, ^15^N, and ^195^Pt NMR Spectroscopy

A series of ^1^H, ^13^C, ^15^N, and ^195^Pt NMR spectra were recorded to confirm the composition of the coordination sphere in a solution and indicate the donor atoms bonded to the platinum(II) ion. As mentioned above, we also calculated NMR spectral shifts by the DFT methodology for their detailed analysis and characteristic structure-spectral trends.

Most characteristic resonance signals in experimental ^1^H NMR spectra are detected in the aromatic region between 8.53 and 8.66 ppm for H2 and 6.93 and 7.79 ppm for H6 (Table 3 and Table 4). Both were deshielded compared to the resonance signals for the corresponding free ligands. For that reason, we estimated a value of coordination shits (Δ^1^H_coord_ = ^1^Hδ_complex_–^1^Hδ_ligand_) for H2 (Δ^1^H_coord_ = 0.06–0.17 ppm) and for H6 (Δ^1^H_coord_ = 0.07–0.26 ppm), which confirm the coordination of triazolopyrimidines to the platinum(II) ion. Additionally, in ^13^C NMR spectra, we observed that the signals of the C2 and C3a atoms were shielded (Δ^13^C2_coord_ = −2.4 to −1.1 ppm, Δ^13^C3a_coord_ = −3.6 to −2.3 ppm), whereas other atoms were deshielded (Table 3 and Table 4). This suggested that a likely place for coordinating heterocycle ligand was a five-membered triazole ring [14]. The ^1^H-^15^N analysis indicated which nitrogen atom is a donor and forms a coordination bond with the platinum(II) ion.

The results reveal that the resonance signals for N1, N4, and N8 show relatively small shifts (Δ^15^N_coord_ = −2.6 to 1.6 ppm), whereas the shielding of the N3 atom is significantly greater (Δ^15^N_coord_ = −73.1 to −64.5 ppm) (Table 3 and Table 4). The biggest shielding for N3 after coordination is observed for complex (**A2**) and (**B2**) with 5-methyl-7-isobutyl-1,2,4-triazolo[1,5-*a*]pyrimidine (Figure 4). Based on these findings, we implied that N3 is the coordination site. Additionally, the ranges of coordination chemical shifts agree with data of previously discussed complexes with the same triazolopyrimidine coordination [15,16,17]. and with parameters obtained by DFT calculations.

In addition, ^1^H NMR spectra of novel complexes exhibited two multiplets for the CH_2_ groups from TMSO in the 4.18–3.63 ppm and 2.45–2.22 ppm ranges and multiplets for phenyl groups from DPSO in the 7.53–7.55 ppm and 8.06–8.13 ppm ranges. These signals were shifted too (compared with spectra of uncoordinated sulfoxides). Therefore, the binding of tetrahydrothiophene-1-oxide or diphenyl sulfoxide to the platinum(II) ion was confirmed.

Unfortunately, it has not been possible to indicate the isomers (*cis* or *trans*) based on ^1^H, ^13^C, and ^15^N NMR studies. The ^195^Pt NMR experiments were necessary. ^195^Pt resonance signals for studied, mixed platinum(II) complexes shift towards higher fields than K_2_PtCl_4,_ and are observed between −3000 and −3010 ppm for compounds with TMSO or between −3095 and −3101 ppm for compounds with DPSO (Figure 5). It indicates that shielding spectral signals for trans isomers of platinum(II) complexes with diphenyl sulfoxide increase compared to *cis* isomers of analogous complexes with tetrahydrothiophene-1-oxide. A similar observation is noted for previously investigated complexes with a dimethyl sulfoxide (DMSO), such as *cis-*[PtCl_2_(DMSO)(ibmtp)] [11] and *trans*-[PtCl_2_(DMSO)(dptp)] [12,18]. Additionally, an occurrence of signals in a very tight range indicates that the replacement of N-donor heterocycle ligand to other triazolopyrimidine derivatives in this type of complexes has a negligible impact on ^195^Pt resonance signal shifts [12]. On the other hand, changing the N-donor ligand to a sulfoxide (DMSO or DPSO) significantly affects the position of the signal of the ^195^Pt NMR spectra (Figure 5). The good agreement of these resonance signals with DFT ^195^Pt chemical shifts suggested that the ^195^Pt NMR experiments can be useful for determining a specific geometric isomer for these complexes.

Potential energy surfaces (PES) for (**B**) isomers, showing the computed DFT energy scan as triazolopyrimidines (dstp) and DPSO ligands rotating about N3-Pt1-S1 *trans*-bonds, are presented in Figure 6, panel (**A**). All four (**B1**–**B4**) isomers give comparable results as the dstp plane turns around the two bulky phenyl groups of a DPSO ligand. The scanned torsional angle (C3a-[N-Pt1-S1]-O1) is between the C3a-N3-Pt1 plane of a dstp ligand and the Pt1-S1=O1 plane of the DPSO ligand. The plots reveal shallow torsional energy barriers, with easily identified four (two pairs) energy minima. Each minimum is nearly eclipsed (0 degrees) or staggered (180 degrees) conformations. The two global minima, tilted only by around ±15 degrees from an eclipsed conformation, place the C3a carbon of the dstp ring and the O1 oxygen of DPSO on the same side of the Pt coordination plane. The other minima pair, also tilted by around ±15 degrees, but from a staggered conformation (at ±165 degrees), places the C3a carbon and O1 oxygen atoms opposite the Pt coordination plane. The two pairs of rotamers, one nearly eclipsed and another nearly staggered, are energetically comparable (0.5 kcal/mol) and separated by two gauche-type transitional conformations (around ±60 degrees and ±120 degrees), and the rotational energy barrier of 1.75 kcal/mol when the O1 atom eclipses one of the chlorides. Therefore, the tp rings have complete rotational freedom about the DPSO ligand at room temperature. Notably, in these complexes′ crystal structures, any rotational configuration of the tps plane seems plausible, especially in the presence of favorable intermolecular interactions (not included in the calculations).

Potentially, the dstp plane′s rotational freedom, illustrated for (**B**) complexes, might significantly obstruct NMR aid to support the characterization of the Pt-N3 bond. We have also studied H, C, N, and Pt NMR shifts as a function of the C3a-[N3-Pt1-S1]-O1 torsional angle, as defined above. The calculations revealed that only ^195^Pt NMR shifts are appreciably sensitive to the rotational angle, altering the computed value by about 250 ppm, ranging from −3500 to −3250 ppm. The dot-plot in panel B of Figure 6 illustrates these results.

For most stable rotamers of the (**B**) complexes, as discussed above, two nearly eclipsed (0 degrees) and another two nearly staggered (180 degrees) conformations, the computed ^195^Pt NMR shifts are at around −3450 ppm, while, experimentally, the signal is observed at around −3100 ppm. The discrepancy between computed and experimental values found here is more prominent than that found for other Pt(II) complexes. However, given these complexes′ conformational accessibility and rotational freedom, one should consider a complete range of computed ^195^Pt NMR shifts (from −3500 to −3250 ppm) as very plausible. Thus, the ^195^Pt NMR shift marker for (**B**) complexes and those structurally similar should be applied cautiously, as it presents a challenge for the definite characterization of such complexes.

In summary of the section, the relationship between the rotational isomerization induced by the DPSO ligand and ^195^Pt NMR shifts found via computational modeling revealed subtle and essential differences. Within the full 360-degree rotation of the DPSO ligand, the computed ^195^Pt NMR shifts for *trans* and *cis* isomers range from −3100 to −3500 ppm (Figure 7). The lower region of the spectral gap between −3375 and −3500 ppm might be deterministic for the *trans* isomers with the torsional angle near the eclipse conformation at 0 degrees (within +/− 60 degrees). Moreover, the ^195^Pt NMR shifts in the higher spectral region, between −3100 and −3225 ppm, might easily characterize the *cis* isomers with the torsional angle near their minima (within +/−20 degrees). However, the computed spectral signals between −3375 and −3225 ppm are representative for both *trans* and *cis*-isomers (Figure 7). Therefore, the isomer determination of *cis* versus *trans* based on these spectral lines is inconclusive.

## 3. Materials and Methods

### 3.1. Materials and Instrumentation

The reagents: 3-amino-1,2,4-triazole, acetylacetone, 1,3-diphenyl-1,3-propanedione, 2,2,6,6-tetramethyl-3,5-heptanedione, 6-methyl-2,4-heptanedione, diphenyl sulfoxide, tetrahydrothiophene-1-oxide, and K_2_PtCl_4_ were purchased from Sigma Aldrich (St. Louis, MO, United States), whereas the inorganic salts and solvents of analytical grade from Avantor Performance Materials Poland S.A. (Gliwice, Poland). All chemicals and solvents obtained commercially were used without further purification.

The 5,7-disubstituted derivatives of 1,2,4-triazolo[1,5-*a*]pyrimidine were obtained according to the Bülow and Haas [19,20] method by the condensation of 3-amino-1,2,4-triazole with corresponding diketones such as: acetylacetone for (dmtp), 6-methyl-2,4-heptanedione for (ibmtp), 2,2,6,6-tetramethyl-3,5-heptanedione for (dbtp) and 3,5-diphenyl-1,3-propanedione for (dptp).

### 3.2. Instrumentation

The C, H, and N elemental contents were determined using an Elementar Analysensysteme GmbH Vario MACRO analyzer. ATR-FTIR spectra of free ligands and novel platinum(II) complexes were recorded using Vertec 70v spectrometer (200–4000 cm^−1^). Moreover, the UV-Vis measurements for lipophilicity determination were obtained using a HITACHI U-2900 UV-Vis spectrophotometer equipped with 1.0 cm path length quartz cuvettes (1.5 mL).

NMR spectra were recorded at 298 K in the CDCl_3_ solutions with a Varian INOVA 500 (Varian Inc., Palo Alto, CA, USA) spectrometer, operating at 499.8, 125.7, 50.6, and 107.4 MHz for ^1^H, ^13^C, ^15^N, and ^195^Pt, respectively. The reference standards were TMS for ^1^H and ^13^C, CH_3_NO_2_ for ^15^N, and H_2_PtCl_6_ for ^195^Pt. Gradient-enhanced IMPACT-HMBC ^1^H-{^15^N} correlation spectra [21] were optimized for the coupling constant of 6 Hz under the following experimental conditions: an acquisition time of 0.2 s, spectral windows of 6000 (F2) and 10,000 (F1) Hz, 1K complex data points, 256 time increments, 30 ms WURST-2 mixing sequence centered within a 60 ms preparation interval (ASAP^2^) and a 150° Ernst angle as the excitation pulse [22].

### 3.3. X-Ray Structure Determination

Crystals of (**A1**, **A3**, **B3**) were grown from the ethanol solution. The X-ray diffraction data were collected with an Oxford Sapphire CCD diffractometer using MoKα radiation, λ = 0.71073 Å, by ω-2θ method at 293 (2) K. Structures were solved by the direct methods and refined with the full-matrix least-squares method on F2 using SHELX2017 program package [23,24]. Analytical absorption corrections were applied (CrysAlis package of programs [25]) (Table 5). In all structures, hydrogen atoms were located from the difference of electron density maps. Their positions were constrained in the refinement with the appropriate AFIX commands as implemented in SHELX. The structural data of (**A1**, **A3**, **B3**) are deposited with Cambridge Crystallographic Data Centre, CCDC number 2142681 (**A1**), 2142682 (**A3**), and 2142683 (**B3**).

### 3.4. Computational Methods

Density functional theory (DFT) was employed to compute geometry, frequencies, and NMR shifts of the eight synthesized Pt(II) complexes (**A1**–**A4**, **B1**–**B4**) as implemented in Gaussian 09 software package [26]. Moreover, an alternative series of eight complexes that are geometric isomers of (**A**) and (**B**), i.e., *trans*- for (**A**) and *cis*- for (**B**), were included in the DFT computations to deepen the energetic and spectroscopic comparisons. Specifically, geometry optimization for all structures was performed with B3LYP functional and 6–311G** basis set for all atoms but Pt atom, where aug-cc-pVTZ-PP basis sets [27] were applied. All frequencies of optimized structures were positive, indicating that their geometry converged to appropriate minima. Interestingly, the relative orientation of the tp ligand’s aromatic ring, in all optimized structures, is always perpendicular to the plane of the Pt(II) coordination sphere, PtCl_2_(N3-tp) (S1), where the C3a atom is pointing away from the metal′s coordinated sulfur atom (S1).

The NMR chemical shifts of ^1^H, ^13^C, ^15^N, and ^195^Pt were computed with gauge-including atomic orbitals (GIAO) method [28,29] using B3LYP functional with SARC-ZORA basis sets for Pt atom and 6–311++G** basis set for the remaining atoms (GIAO-B3LYP/SARC-ZORA/6–311++G**). All basis sets were obtained from the Basis Set Exchange Library [30]. Moreover, a solvent environment was represented applying the polarizable continuum model (PCM) [31] with a polarizable solvent cavity (ε = 4.7113), consistently for all modeled structures and their spectral predictions. The computational procedure outlined here has proven successful in our previous parent Pt(II) complex studies [11]. Interestingly, the (**B**) complexes containing a bulky DPSO ligand show several distinct minima along a torsional angle of the Pt-S bond, defined by O1-[S1-Pt1-N3]-C3a atom sequence. More importantly, the computed ^195^Pt NMR chemical shift along the torsional angle shows significant fluctuation, ranging as much as 250 ppm. At the same time, the rotational energy barrier is very shallow, only about 1.75 kcal/mole. Consequently, the ^195^Pt NMR shifts for (**B**) complexes are not as distinct or conclusive as series A complexes. Computed relative energies (kcal/mol) and the ^195^Pt NMR shifts (ppm) as a function of the torsion of the DPSO ligand for (**B1**–**B4**) complexes and their complementing *cis*-isomers are listed in the table given in the Appendix A.

### 3.5. Syntheses of Complexes

*cis*-[PtCl_2_(TMSO)_2_] were used as a substrate for syntheses of (**A1**–**A4**), and *cis*-[PtCl_2_(DPSO)_2_] were used as a substrate for syntheses of (**B1**–**B4**). Starting complexes were prepared from K_2_PtCl_4_ and tetrahydrothiophene-1-oxide or diphenyl sulfoxide by the method analogous to that for *cis*-[PtCl_2_(DMSO)_2_] [32]. The details of these syntheses are as follows:

#### 3.5.1. cis-[PtCl_2_(TMSO)_2_]

A solution of K_2_PtCl_4_ (0.100 g; 0.24 mmol) in 10 mL of water was treated with TMSO (48 μL; 0.53 mmol). The reaction mixture was stirred at room temperature for 24 h. The white participant was filtered, washed with water, ethanol, diethyl ether, and dried under vacuum.

Yield 0.092 g (80%). ^1^H NMR (499.8 MHz, CDCl_3_): δ = 2.23 (m, 2H, -CH_2_), 2.40 (m, 2H, CH_2_), 3.55 (m, 2H, S-CH_2_), 4.17 (m, 2H, S-CH_2_); ^13^C NMR (125.7 MHz, CDCl_3_): δ = 25.4 (CH_2_), 58.6 (S-CH_2_); ^195^Pt NMR (107.4 MHz, CDCl_3_): −3436 ppm (s, PtCl_2_S_2_).

Elemental analysis found C, 20.2; H, 3.3%. Calculation for C_8_C_l2_H_16_O_2_PtS_2_: C, 20.3; H, 3.4%.

#### 3.5.2. *cis*-[PtCl_2_(DPSO)_2_]

A solution of K_2_PtCl_4_ (0.100 g; 0.24 mmol) in 10 mL of water was treated with an equivalent of DPSO (0.097 g, 0.48 mmol) in 5 mL of acetone. The reaction mixture was stirred at room temperature for 48 h. Slow evaporation of solvents obtained a white precipitate, which was filtered, washed with diethyl ether, and dried under vacuum.

Yield 0.082 g (51%). ^1^H NMR (499.8 MHz, CDCl_3_): δ = 7.50 (m, 4H, o-C_6_H_5_), 2.40 (m, 2H, p-C_6_H_5_), 7.85 (m, 4H, m-C_6_H_5_); ^13^C NMR (125.7 MHz, CDCl_3_): δ = 127.7 (o-C_6_H_5_), 128.8 (m-C_6_H_5_), 133.2 (p-C_6_H_5_), 140.8 (S-C_6_H_5_); ^195^Pt NMR (107.4 MHz, CDCl_3_): −3529 ppm (s, PtCl_2_S_2_).

Elemental analysis found C, 42.8; H, 3.4%. Calculation for C_24_Cl_2_H_20_O_2_PtS_2_: C, 43.0; H, 3.0%.

#### 3.5.3. cis-[PtCl_2_(dstp) (TMSO)] (**A1**–**A4**) and trans-[PtCl_2_(dstp) (DPSO] (**B1**–**B4**)

All novel platinum(II) complexes with tetrahydrothiophene-1-oxide were obtained by reaction between starting complex *cis*-[PtCl_2_(TMSO)_2_] or *cis*-[PtCl_2_(DPSO)_2_] and corresponding dstp ligands (dmtp (**A1**, **B1**), ibmtp (**A2**, **B2**), dbtp (**A3**, **B3**) and dptp (**A4**, **B4**)) in molar ratios of 1:1 in ethanol. The reaction mixture was heated to 45 °C and stirred for 2 h. After this period, stirring continued at r.t. for 24 h. Slow evaporation of the solvent resulted in obtaining yellow residue, which was freeze-dried (**A1**–**A3**, **B1**–**B3**). A deposit formed directly in the reaction mixture (**A4**, **B4**) was washed with diethyl ether and then dried under a vacuum.

Yield 65% (**A1**), 68% (**B1**), 70% (**A2**), 62% (**B2**), 71% (**A3**), 57% (**B3**), 75% (**A4**), 79% ^1^H. Analysis calc/found for: (**A1**) C_11_Cl_2_H_16_N_4_OPtS: C, 25.5/25.2%; H 3.1/3.6%; N, 10.8/10.8%; (**B1**) C_19_Cl_2_H_18_N_4_OPtS: C, 37.0/37.0%; H 2.9/2.8%; N, 9.1/9.0%; (**A2**) C_14_Cl_2_H_22_N_4_OPtS: C, 30.0/29.7%; H, 4.0/3.9%; N, 9.9/9.4%; (**B2**) C_22_Cl_2_H_24_N_4_OPtS: C, 40.1/39.9%; H, 3.7/4.3%; N, 8.5/8.4%; (**A3**) C_17_Cl_2_H_28_N_4_OPtS: C, 33.9/34.0%; H, 4.7/4.7%; N, 9.3/8.7 %; (**B3**) C_25_Cl_2_H_30_N_4_OPtS: C, 42.9/43.0%; H, 4.3/4.7%; N, 8.0/7.9 %; (**A4**) C_21_Cl_2_H_20_N_4_OPtS: C, 39.2/39.5%; H, 3.1/3.4%; N, 8.7/8.5%; (**B4**) C_29_C_l2_H_22_N_4_OPtS: C, 47.0/47.2%; H, 3.0/3.5%; N, 7.6/7.6%.

## 4. Conclusions

We successfully synthesized eight novel square planar platinum(II) complexes with 5,7-disubstituted-1,2,4-triazolo[1,5-*a*]pyrimidine (dstp) and two different sulfoxides: DPSO or TMSO. The structures, and the effects of sulfoxide ligands on their *cis*-*trans* isomerism, were studied by employing multinuclear (^1^H, ^13^C, ^15^N, ^195^Pt) NMR and X-ray experiments and DFT calculations. Based on a good agreement of all experimental results with DFT calculations, we suggest that ^195^Pt NMR (experiments and DFT parameters) is a valuable method for predicting the *cis-trans* isomerism of platinum(II) complexes without the crystal structure.

## Figures and Tables

**Figure 1 ijms-23-03656-f001:**
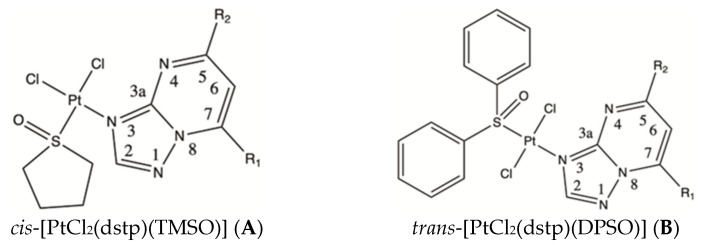
Structures of novel dichloride platinum(II) complexes. The structures contain tetrahydrothiophene-1-oxide (TMSO) for *cis* isomers (**A**) and diphenyl sulfoxide (DPSO) for *trans* isomers (**B**). IUPAC numbering of 1,2,4-triazolo[1,5-*a*]pyrimidine derivative is displayed. (**A1**, **B1**) R_1_, R_2_ = CH_3_ for dmtp; (**A2**, **B2**) R_1_ = CH_2_CH(CH_3_)_2_, R_2_ = CH_3_ for ibmtp; (**A3**, **B3**) R_1_, R_2_ = C(CH_3_)_3_ for dbtp; (**A4**, **B4**) R_1_, R_2_ = C_6_H_5_ for dptp.

**Figure 2 ijms-23-03656-f002:**
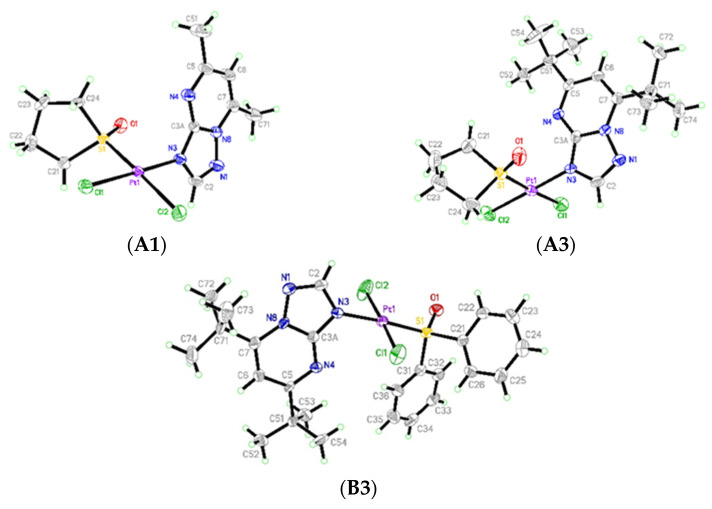
Crystal structure of *cis*-[PtCl_2_(dmtp)(TMSO)] (**A1**), *cis*-[PtCl_2_(dbtp)(TMSO)] (**A3**) and *trans-*[PtCl_2_(dbtp) (DPSO)] (**B3**).

**Figure 3 ijms-23-03656-f003:**
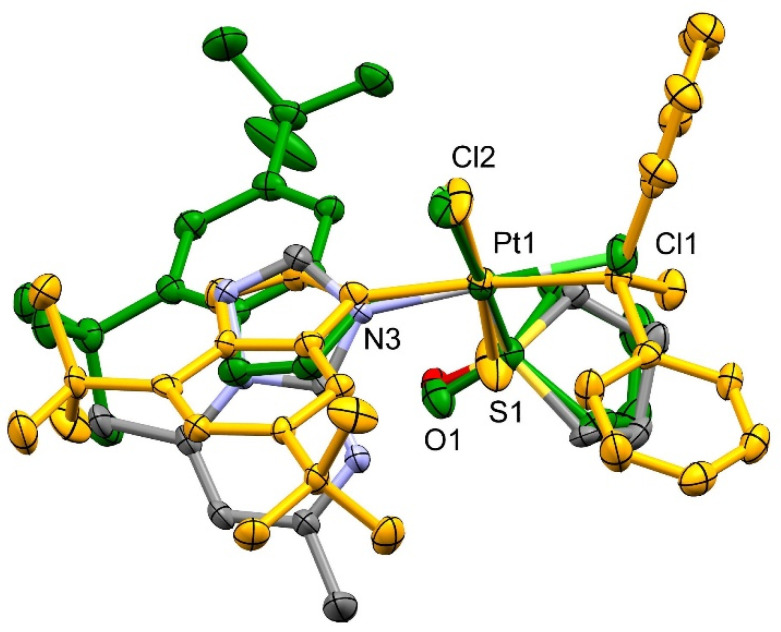
Structures superposition of (**A1**) (atoms: C **gray**, N **blue**, O **red**), (**A3**) (**green**) and (**B3**) (**yellow**). Heteroatoms are labeled for (**A1**).

**Figure 4 ijms-23-03656-f004:**
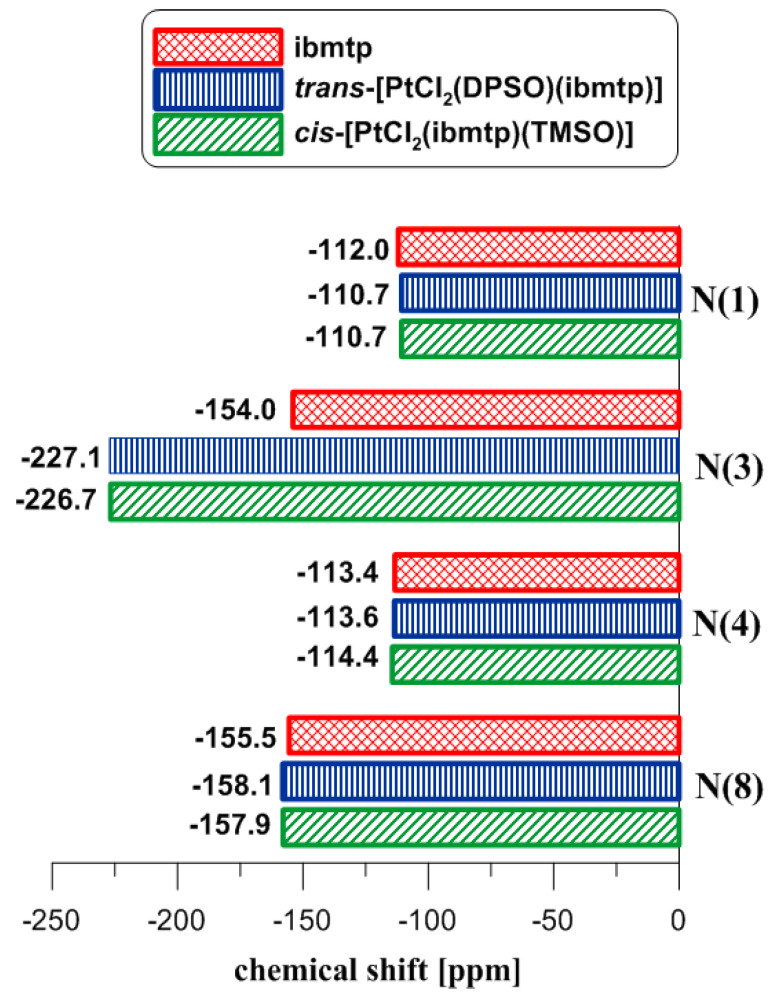
^15^N NMR chemical shifts of free ligand (ibmtp) and platinum(II) complexes *cis-*[PtCl_2_(ibmtp)(TMSO)] and *trans*-[PtCl_2_(DPSO) (ibmtp)].

**Figure 5 ijms-23-03656-f005:**
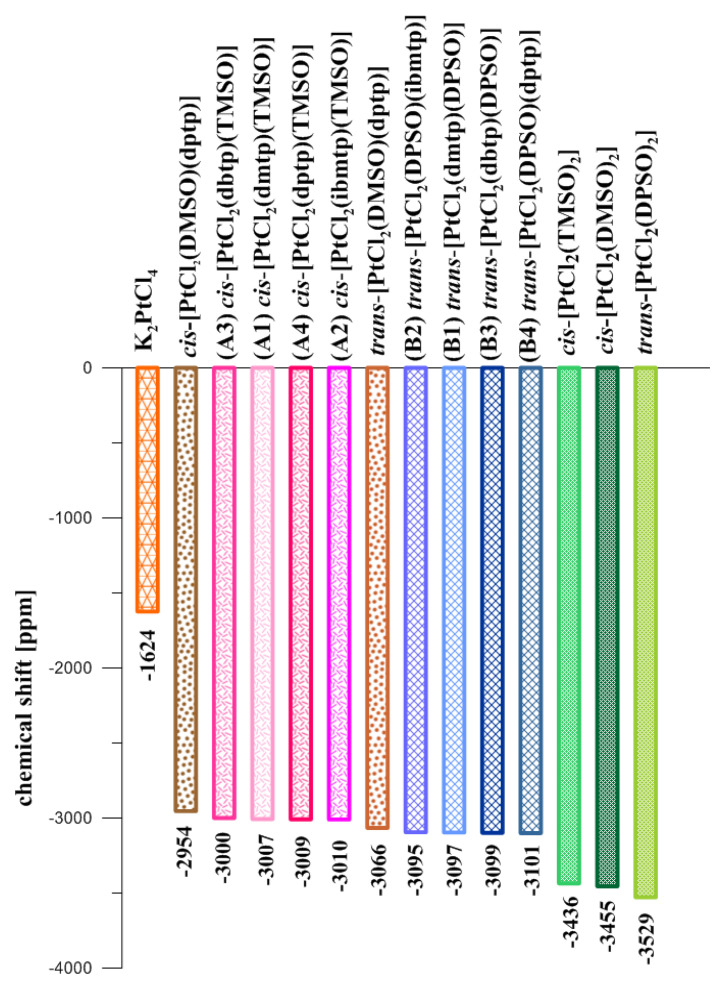
^195^Pt NMR chemical shifts of a substrate (K_2_PtCl_4_) and platinum(II) complexes with sulfoxides.

**Figure 6 ijms-23-03656-f006:**
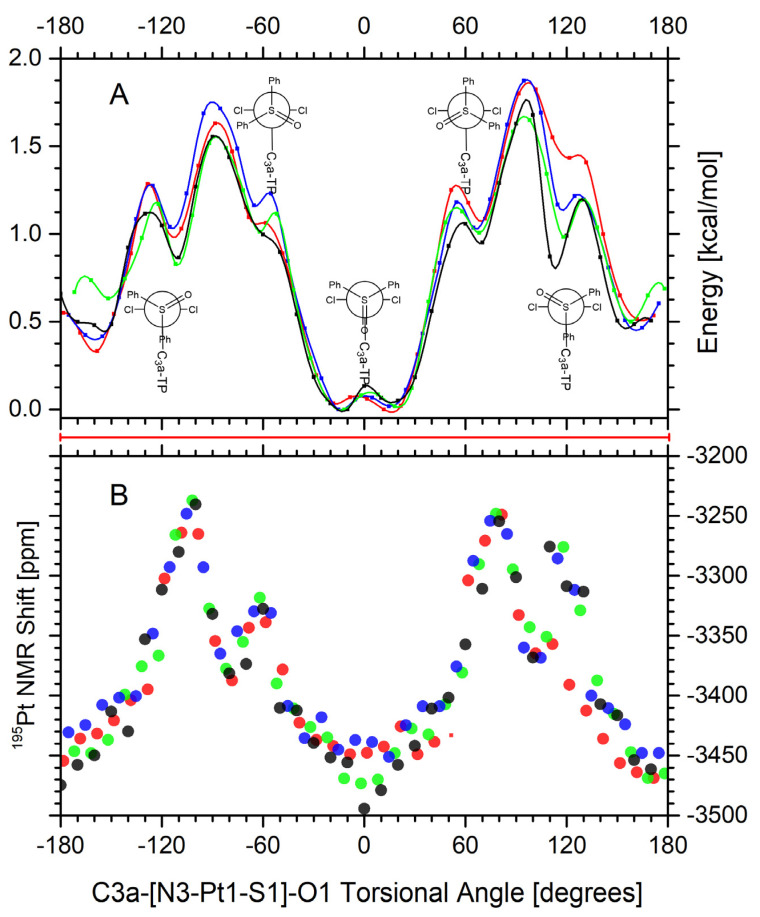
Calculated scan of the potential energy surface (panel (**A**)) and ^195^Pt NMR shifts (panel (**B**)) as a function of the (C3a)-Pt1-S1 = O1 torsional angle for *trans*-isomers of dmtp (**black**), dbtp (**blue**), ibmtp (**green**), and dptp (**red**). A red bar between the panels indicates a degree of rotational freedom within about 2 kcal/mol above the rotational energy minimum (complete 360-degree rotation). See a table in the supplementary material for values used in the graphs.

**Figure 7 ijms-23-03656-f007:**
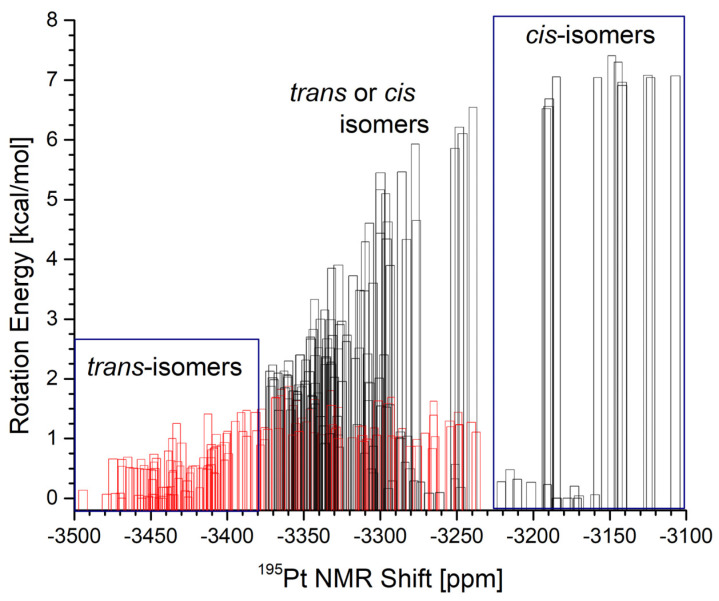
Histogram of calculated ^195^Pt NMR for (**B1**–**B4**) complexes, *trans*-isomers (**red bars**), and corresponding *cis*-isomers (**black bars**) of [PtCl_2_(DPSO)(dstp)] structures. See a table in the supplementary materials for values used in the graphs.

**Table 1 ijms-23-03656-t001:** Experimental and calculated selected bond lengths [Å] and angles [°] for (**A1**), (**A3**), and (**B3**).

	(A1)		(A3)		(B3)	
**Bond Lengths [Å]**						
	Exp.	Calc.	Exp.	Calc.	Exp.	Calc.
Pt1-N3	2.017 (4)	2.043	2.020 (6)	2.046	2.033 (6)	2.052
Pt1-S1	2.199 (14)	2.256	2.199 (2)	2.274	2.215 (18)	2.267
Pt1-Cl1	2.301 (12)	2.343	2.307 (2)	2.349	2.281 (2)	2.357
Pt1-Cl2	2.306 (15)	2.351	2.299 (2)	2.342	2.295 (2)	2.349
**Angles [°]**						
	Exp.	Calc.	Exp.	Calc.	Exp.	Calc.
N3-Pt1-S1	89.65 (13)	91.5	89.28 (18)	89.33	175.29 (16)	175.6
N3-Pt1-Cl1	178.63 (11)	178.7	88.55 (19)	87.47	88.28 (18)	87.6
S1-Pt1-Cl1	90.60 (5)	91.5	177.30 (8)	176.28	92.86 (7)	94.5
N3-Pt1-Cl2	88.56 (13)	87.9	179.17 (19)	177.19	87.58 (18)	88.0
S1-Pt1-Cl2	175.51 (5)	179.3	90.60 (9)	93.43	91.32 (8)	89.8
Cl1-Pt1-Cl2	91.29 (5)	90.8	91.54 (9)	89.75	175.82 (9)	175.2

**Table 2 ijms-23-03656-t002:** DFT computed values of selected bond lengths (Å) and angles (°) for (**A2**, **A4**, **B1**, **B2**, **B4**).

	(A2)	(A4)	(B1)	(B2)	(B4)
**Bond Lengths [Å]**					
Pt1-N3	2.042	2.043	2.053	2.054	2.052
Pt1-S1	2.259	2.258	2.267	2.266	2.267
Pt1-Cl1	2.351	2.349	2.348	2.350	2.355
Pt1-Cl2	2.341	2.343	2.355	2.355	2.355
**Angles [°]**					
N3-Pt1-S1	91.5	91.4	175.4	175.4	175.7
N3-Pt1-Cl1	87.7	87.8	87.4	87.3	87.8
S1-Pt1-Cl1	178.6	179.1	94.5	94.6	90.5
N3-Pt1-Cl2	178.5	178.5	87.9	88.0	87.7
S1-Pt1-Cl2	89.9	89.9	90.1	90.0	90.5
Cl1-Pt1-Cl2	90.9	90.9	174.8	175.1	174.6

**Table 3 ijms-23-03656-t003:** Experimental and calculated ^1^H, ^13^C, ^15^N, and ^195^Pt NMR chemical shifts for selected atoms of the platinum(II) complexes with tetrahydrothiophene-1-oxide (TMSO).

	(A1)		(A2)		(A3)		(A4)	
	Expt.	Calc.	Expt.	Calc.	Expt.	Calc.	Expt.	Calc.
δH2	8.53 (+0.13)	8.82	8.56 (+0.10)	8.83	8.56 (+0.14)	8.86	8.66 (+0.07)	8.88
δH6	7.07 (+0.26)	7.55	6.95 (+0.22)	7.52	7.16 (+0.16)	7.74	7.79 (+0.07)	8.30
δC2	153.0 (−2.2)	153.10	152.9 (−2.3)	152.72	152.1 (−2.1)	151.56	153.7 (−1.1)	153.79
δC3a	151.9 (−3.2)	151.34	151.8 (−3.6)	151.87	152.5 (−3.4)	151.90	153.0 (−2.3)	152.63
δC5	168.7 (+4.0)	168.86	168.6 (+3.9)	168.21	179.1 (+3.4)	179.50	164.4 (+1.9)	163.00
δC6	113.4 (+2.6)	110.44	113.2 (+2.7)	111.24	106.0 (+2.6)	106.01	108.8 (+1.6)	106.64
δC7	148.1 (+1.4)	149.74	151.1 (+1.4)	152.49	158.7 (+1.3)	160.73	149.3 (+0.9)	150.21
δN1	−110.6 (+1.6)	−116.91	−110.7 (+1.3)	−118.69	−106.6 (+0.4)	−113.26	−110.3 (+1.0)	−117.28
δN3	−226.2 (−72.2)	−207.31	−226.7 (−72.7)	−206.98	−228.0 (−70.9)	−208.19	−225.5 (−64.6)	−207.80
δN4	−114.5 (−1.1)	−125.52	−114.4 (−1.0)	−126.62	−114.7 (+0.2)	−130.09	−122.2 (−0.4)	−137.47
δN8	−157.2 (−2.3)	−165.06	−157.9 (−2.4)	−165.27	−160.3 (−2.2)	−166.95	−162.0(−2.2)	−170.60
δPt	−3010	−3209.32	−3007	−3202.16	−3000	−3186.18	−3009	−3195.93

**Table 4 ijms-23-03656-t004:** Experimental and calculated ^1^H, ^13^C, ^15^N, and ^195^Pt NMR chemical shifts for selected atoms of the platinum(II) complexes with diphenyl sulfoxide (DPSO).

	(B1)		(B2)		(B3)		(B4)	
	Expt.	Calc.	Expt.	Calc.	Expt.	Calc.	Expt.	Calc.
δH2	8.57 (+0.17)	8.87	8.55 (+0.09)	8.86	8.54 (+0.12)	8.92	8.65 (+0.06)	8.97
δH6	7.01 (+0.20)	7.57	6.93 (+0.20)	7.55	7.15 (+0.15)	7.74	7.79 (+0.07)	8.36
δC2	152.9 (−2.3)	152.81	152.8 (−2.4)	152.69	151.8 (−2.4)	152.13	153.6 (−1.2)	153.59
δC3a	151.8 (−3.3)	151.62	152.1 (−3.3)	151.72	152.5 (−3.4)	152.33	152.9 (−2.4)	152.77
δC5	168.8 (+4.1)	168.44	168.6 (+3.9)	168.37	179.0 (+3.3)	180.05	164.2 (+1.7)	163.33
δC6	113.4 (+2.6)	110.50	113.1 (+2.6)	111.24	106.0 (+2.6)	103.99	108.7 (+1.5)	106.82
δC7	148.0 (+1.3)	149.77	151.0 (+1.3)	152.67	158.7 (+1.3)	161.84	149.3 (+0.9)	150.65
δN1	−110.6 (+1.6)	−117.15	−110.7 (+1.3)	−117.15	−106.0 (+1.0)	−113.19	−110.0 (+1.3)	−117.58
δN3	−226.6 (−72.6)	−213.13	−227.1 (−73.1)	−213.80	−227.7 (−70.6)	−217.63	−225.4 (−64.5)	−213.41
δN4	−113.7(−0.3)	−126.29	−113.6 (−0.2)	−126.25	−115.1 (−0.2)	−125.87	−122.0 (+0.2)	−136.29
δN8	−157.4 (−2.5)	−163.81	−158.1 (−2.6)	−164.62	−160.4 (−2.3)	165.46	−162.0 (−2.2)	169.46
δPt	−3097	−3465.9	−3095	−3466.9	−3099	3445.3	−3101	3442.8

**Table 5 ijms-23-03656-t005:** Crystal data and structure refinement for (**A1**, **A3** and **B3**).

Compound	*cis*-[PtCl_2_(dmtp)(TMSO)] (A1)	*cis*-[PtCl_2_(dbtp)(TMSO)] (A3)	*trans*-[PtCl_2_(dbtp (DPSO)] (B3)
Empirical formula	C_11_H_16_Cl_2_N_4_OPtS	C_17_H_28_Cl_2_N_4_OPtS	C_25_H_30_Cl_2_N_4_OPtS
Formula weight	518.33	602.48	700.58
Temperature; K	293 (2)	293 (2)	293 (2)
Wavelength; Å	0.71073	0.71073	0.71073
Crystal system	Monoclinic	Monoclinic	Triclinic
Space group	P2_1_/n	P2_1_/n	P-1
Unit cell dimensions; Å, °	a = 10.7446 (3)	a = 12.3610 (16)	a = 9.3248 (7)
	b = 8.3993 (3)	b = 10.1722 (10)	b = 10.8972 (8)
	c = 18.0425 (6)	c = 17.871 (2)	c = 14.6506 (12)
	α = 90	α =90	α =88.183 (6)
	β = 98.946 (3)	β = 94.347 (10)	β = 73.143 (7)
	γ = 90	γ = 90	γ = 77.059 (6)
Volume; Å^3^	1608.48 (10)	2240.6 (5)	1387.67 (19)
Z	4	4	2
Density (calculated); Mg/m^3^	2.140	1.786	1.677
Absorption coefficient; mm^−1^	9.185	6.608	5.348
F (000)	984	1176	688
Crystal size; mm	0.340 × 0.209 × 0.136	0.609 × 0.579 × 0.194	0.312 × 0.237 × 0.086
Theta range for data collection	2.285 to 28.462°.	2.305 to 28.486°.	2.342 to 28.516°.
Index ranges	−13 < = h < = 14, −10 < = k < = 10, −23 < = l < = 21	−15 < = h < = 15, −12 < = k < = 13, −21 < = l < = 23	−12 < = h < = 12, −13 < = k < = 13, −19 < = l < = 13
Reflections collected	10,499	15,074	9764
Independent reflections	3696 [R(int) = 0.0474]	5184 [R(int) = 0.0716]	6129 [R(int) = 0.0389]
Completeness to theta = 25.242°	100.0 %	99.9 %	99.9 %
Max. and min. transmission	0.429 and 0.135	0.377 and 0.060	0.657 and 0.366
Refinement method	Full-matrix least-squares on F^2^	Full-matrix least-squares on F^2^	Full-matrix least-squares on F^2^
Data/restraints/parameters	3696/0/181	5184/0/235	6129/0/307
Goodness-of-fit on F^2^	0.995	1.052	1.063
Final R indices [I > 2sigma(I)]	R_1_ = 0.0314, wR_2_ = 0.0497	R_1_ = 0.0541, wR_2_ = 0.1178	R_1_ = 0.0412, wR_2_ = 0.0744
R indices (all data)	R_1_ = 0.0524, wR_2_ = 0.0539	R_1_ = 0.0816, wR_2_ = 0.1347	R_1_ = 0.0597, wR_2_ = 0.0918
Largest diff. peak and hole; e.Å^−3^	1.203 and −0.791	2.862 and −1.726	d −0.768

## Data Availability

Faculty of Chemistry, Nicolaus Copernicus University in Toruń.

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
