# Peer review of "Factors Affecting the Stability of Platinum(II) Complexes with 1,2,4-Triazolo[1,5-a]pyrimidine Derivatives and Tetrahydrothiophene-1-Oxide or Diphenyl Sulfoxide"

_ijms, 2022, doi:10.3390/ijms23073656_

Round 1

Reviewer 1 Report

The present paper reports the synthesis of new series of chlorido Pt(II) complexes and their characterization by the employment of X-ray diffraction, multinuclear NMR spectroscopy and DFT calculations. In particular, the possibility to distinguish cis and trans isomers has been examined in depth. The manuscript is properly organized and the content well presented. Nevertheless, the authors do not give any information about the possible use of such complexes and the aim for which they have been synthesized. Theoretical calculations should be used, for example, to predict the behavior of the complexes with respect to the use for which they have been designed. Synthetic challenges are not enough for the present journal. Eventually, the manuscript might be submitted to a more specific journal.

Author Response

REF. 1.

The present paper reports the synthesis of new series of chlorido Pt(II) complexes and their characterization by the employment of X-ray diffraction, multinuclear NMR spectroscopy and DFT calculations. In particular, the possibility to distinguish cis and trans isomers has been examined in depth. The manuscript is properly organized and the content well presented. Nevertheless, the authors do not give any information about the possible use of such complexes and the aim for which they have been synthesized. Theoretical calculations should be used, for example, to predict the behavior of the complexes with respect to the use for which they have been designed. Synthetic challenges are not enough for the present journal. Eventually, the manuscript might be submitted to a more specific journal.

In the introduction, we added information about antitumor activities of dichloride platinum(II) complexes.

Reviewer 2 Report

The authors present a comprehensive and detailed study of the structure of novel Pt(II) complexes based on multinuclear NMR spectroscopy, DFT calculations and X-ray crystal structure determinations. The work is carefully executed and results and discussion are clearly described. Also the reasons explaining changes in geometrical parameters among different complexes are convincing. I have only minor comments that the authors may take into consideration:

Line 39: please check the formula trans-[PtCl2(tu)2]Cl2

Line 89: … are significantly longer… ?

Figure 2. Atom numbering is hard to read

Line 109: … A1(gray) …

In Figure 3 there is an O atom in red, different from the remaining structures

Line 151: this is a remarkable finding if DFT calculations refer to an isolated complex in the gas phase

Line 157: … as single crystals…?

Line 212: negligible rather than neglected

Line 344: Sentence starting with The analysis of the potential energy surfaces … appears to be misplaced here

The reader may wonder whether upon increasing steric hindrance in the ligation sphere around the metal, sulfoxide coordination may shift from S-Pt to O-Pt. Maybe the authors can comment about this possibility? Or calculate the relative stability?

The authors should choose between sulphoxide and sulfoxide notation and be consistent throughout the manuscript

The interactions named by the authors “hydrophobic attractive interactions” are better classified as “dispersion attractive interactions”

The results of DFT calculations regarding relative energies of isomers could be provided in a table rather than given a qualitative reference to. Are calculations run on isolated species or in a solvent model? In the experimental section PCM is applied for calculations of chemical shifts.

The English language might be improved in some places

Author Response

REF. 2.

Comments and Suggestions for Authors

The authors present a comprehensive and detailed study of the structure of novel Pt(II) complexes based on multinuclear NMR spectroscopy, DFT calculations and X-ray crystal structure determinations. The work is carefully executed and results and discussion are clearly described. Also the reasons explaining changes in geometrical parameters among different complexes are convincing. I have only minor comments that the authors may take into consideration:

Line 39: please check the formula trans-[PtCl2(tu)2]Cl2

We corrected formula

Line 89: … are significantly longer… ?

We corrected the sentence

It has been corrected.

Figure 2. Atom numbering is hard to read

In our opinion atom numbering is OK;

Line 109: … A1(gray) …

In Figure 3 there is an O atom in red, different from the remaining structures

The figure caption is improved to clarify the atom’s coloring.

Line 151: this is a remarkable finding if DFT calculations refer to an isolated complex in the gas phase

All computed structures and the chemical shifts include the solvent effect.

Line 157: … as single crystals…?

Line 212: negligible rather than neglected

We corrected sentences.

Line 344: Sentence starting with The analysis of the potential energy surfaces … appears to be misplaced here

We have replaced the original sentence, and now it says; “Computed relative energies (kcal/mol) and 195Pt NMR shifts (ppm) as a function of the torsion of the DPSO ligand for (B1-B4) complexes and their complementing cis-isomers are listed in the table given in the supplementary materials.

The reader may wonder whether upon increasing steric hindrance in the ligation sphere around the metal, sulfoxide coordination may shift from S-Pt to O-Pt. Maybe the authors can comment about this possibility? Or calculate the relative stability?

It is an excellent point. However, we had not considered the possibility of Pt-O coordination in our calculations. Indeed, it might be considered and addressed if relevant in future investigations of complexes with sulfoxides.

The authors should choose between sulphoxide and sulfoxide notation and be consistent throughout the manuscript

The interactions named by the authors “hydrophobic attractive interactions” are better classified as “dispersion attractive interactions”

The results of DFT calculations regarding relative energies of isomers could be provided in a table rather than given a qualitative reference to. Are calculations run on isolated species or in a solvent model? In the experimental section PCM is applied for calculations of chemical shifts.

A complete table of calculated data used to prepare graphs in figures 6 and 7 are included in the supplementary material.

Also, we clarified in the text that all parts of calculations used the DFT with solvent effects corrected by PCM, i.e., geometry optimization and spectroscopy.

The English language might be improved in some places

I suggested the language is improved throughout the manuscript for clarity of the presentation.

Round 2

Reviewer 1 Report

The authors, to my remarks, answered only adding a statement concerning the possible employment of the synthesized complexes. Beside the fact that instead of "the position of the ligand in the coordination sphere creates of specific properties" it should be "the position of the ligands in the coordination sphere determines the specific properties". What does mean "for their structural .. application"? The aim for carrying out the synthesis continues to be unclear.

Author Response

REF. 1.

The present paper reports the synthesis of new series of chlorido Pt(II) complexes and their characterization by the employment of X-ray diffraction, multinuclear NMR spectroscopy and DFT calculations. In particular, the possibility to distinguish cis and trans isomers has been examined in depth. The manuscript is properly organized and the content well presented. Nevertheless, the authors do not give any information about the possible use of such complexes and the aim for which they have been synthesized. Theoretical calculations should be used, for example, to predict the behavior of the complexes with respect to the use for which they have been designed. Synthetic challenges are not enough for the present journal. Eventually, the manuscript might be submitted to a more specific journal.

Thank you so much for all your suggestions. I would like to inform you that the studies presented here continue our platinum(II) investigation with 5,7-disubstitute-1,2,4-triazolo[1,5-a]pyrimidine compounds as potential anticancer agents. 

From the historical point of view, cis‑diamminedichloridoplatinum(II) was the first platinum-based anticancer agent. It has been in use until now to treat various types of humans. Unfortunately, side effects resulting from its toxicity limit the dose administered to patients. Overcoming these clinical drawbacks in cisplatin-based chemotherapy poses a challenge in developing more effective and less toxic platinum-based anticancer drugs. One of the methods of designing platinum(II) compounds is to modify the pharmacokinetics of cisplatin by replacing the stable ammine ligands with other non leaving N-donor groups. It is well known that ammine ligands substitution with other N‑donor ligands such as 5,7-disubstitiuted-1,2,4-triazolo[1,5-a]pyrimidines which indicate a structure similar to that of the purine bases, could bump the Pt(II) compounds cytotoxicity up. For such complexes are challenging to obtain crystal suitable for X-ray study, but determining a specific isomer of platinum(II) compounds [PtX2N2] is significant for their structural and future application. Therefore we decide to correlate the experimental results with DFT calculations and suggest spectroscopic techniques as a helpful method for predicting the cis/trans isomerism of platinum(II) complexes. 

Round 3

Reviewer 1 Report

The authors have tried to put the outcomes of their work in a context according to reviewer's suggestions.

The paper is now suitable to be published in IJMS.